# “Dr. Google, I am in Pain”—Global Internet Searches Associated with Pain: A Retrospective Analysis of Google Trends Data

**DOI:** 10.3390/ijerph17030954

**Published:** 2020-02-04

**Authors:** Mikołaj Kamiński, Igor Łoniewski, Wojciech Marlicz

**Affiliations:** 1Sanprobi Sp.z.o.o. Sp.K., 70-535 Szczecin, Poland; 2Faculty of Medicine I, Poznan University of Medical Sciences, 60-780 Poznan, Poland; 3Department of Biochemistry and Human Nutrition, Pomeranian Medical University, 70-204 Szczecin, Poland; sanprobi@sanprobi.pl; 4Department of Gastroenterology, Pomeranian Medical University, 70-204 Szczecin, Poland; marlicz@hotmail.com

**Keywords:** Google Trends, Internet, pain, headache, location, ranking, Abdominal pain, back pain, toothache, knee pain

## Abstract

We aimed to rank the most common locations of pain among Google users globally and locally and analyze secular and seasonal trends in pain-related searches in the years 2004–2019. We used data generated by Google Trends (GT) to identify and analyze global interest in topics (*n* = 24) related to locations of pain and how these progressed over time. We analyzed secular trends and time series decomposition to identify seasonal variations. We also calculated the interest in all topics with reference to the relative search volume (RSV) of “Abdominal pain”. Google users were most commonly interested in “Headache” (1.30 [times more frequently than “Abdominal pain”]), “Abdominal pain” (1.00), and “Back pain” (0.84). “Headache” was the most frequent search term in *n* = 41 countries, while “Abdominal pain” was the most frequent term in *n* = 27 countries. The interest in all pain-related topics except “Dyspareunia” increased over time. The sharpest increase was observed for “Abdominal pain” (5.67 RSV/year), and “Toothache” (5.52 RSV/year). Most of the topics revealed seasonal variations. Among pain-related topics, “Headache,” “Abdominal pain,” and “Back pain” interested most Google users. GT is a novel tool that allows retrospective investigation of complaints among Internet users.

## 1. Introduction

Pain is the major reason why people visit a medical office [1]. Up to 60% of individuals may experience pain each month, regardless of age and sex [2], and 22% of primary care patients have been reported to suffer from chronic pain [3]. Globally, Low back pain and migraine are the two leading causes of years lived in disability [4], and pain-related treatment costs are approximately 3% of the gross domestic product (GDP) per capita, which is more than the costs related to cancer and cardiovascular treatment [2]. In addition, 5 billion people do not have access to the opioid analgesics required to treat severe pain [5,6], which underlines the epidemiology of untreated pain [7]. 

In this context, access to the Internet has created new opportunities for patients seeking pain relief. Individuals with chronic conditions predominantly search for health-related information in comparison to people without health problems [8], and many patients with chronic pain administer self-care instead of seeking professional help. These individuals frequently rely solely on the advice of primary care providers, limiting the generation of related data in official health statistics. Therefore, in comparison with research based on health care records, population-based surveys may present more realistic data on the prevalence of pain [9]. In addition, 80% of Internet users perceive Web health-related information as credible [10]. These users can also share their complaints and experiences with peers using forums or social media [11] and quickly find over-the-counter remedies or participate in self-management programs that may provide pain relief [12].

Infodemiology (information epidemiology) is a novel Internet-based approach for analyzing epidemiological data that may not appear in classic epidemiological studies [13,14] For instance, Internet users may experience moderate pain and search for relief on the Web instead of visiting a doctor. Therefore, analysis of Internet data may reveal hitherto poorly investigated relationships. Moreover, infodemiology allows for faster access to data in comparison to standard epidemiological studies, which might have social and economic impact in all fields of medicine dealing with pain. The main sources of infodemiological data are Twitter, Wikipedia, and Google [15].

Globally, Google is the most popular search engine [16], and search engine queries may reflect the actual problems experienced by Web users. The Internet traffic of Google users can be analyzed using Google Trends (GT), which has been used to investigate user interest in Headache [17,18], Toothache [19], and foot and ankle pain [20]. However, to our knowledge, no study has compared the global user interest in pain-related information using Google. This kind of research may unravel the location of pain that mainly bothers Google users. Moreover, the regional differences might be useful for local authorities and scientists to detect the unmet health needs of the local population. Therefore, we aimed to rank the most common body locations of pain among Google users globally and regionally and analyze secular and seasonal trends in pain-related searches in the years 2004–2019.

## 2. Materials and Methods 

### 2.1. Data Collection

GT is a freely accessible tool that enables estimation of the relative search volume (RSV) of a chosen phrase by Google search engine users in a selected region and time since January 2004 (https://trends.google.com/trends/). RSV is an index of search volume adjusted to the number of Google users in a given geographical area. The RSV value ranges from 0 to 100, with a value of 100 indicating the peak of popularity (100% popularity in given period and location) and 0 indicating the nadir (0% popularity) [21]. GT also allows simultaneous comparison of up to five terms, in which an RSV of 100 represents the highest popularity of one of the chosen phrases. GT distinguishes between “search term” and “topic.” Search terms are specific phrases literally typed into the engine, while topics are proposed by GT when typing a specific phrase. Topics enable easy comparisons of given terms across specific regions. For instance, the search term “dog” will be analyzed by GT literally and the RSV will be the highest in English-speaking countries, while the topic “dog” will include all associated queries in all available languages. Therefore, matching topics in English is equivalent to have the option of all languages activated. GT does not count duplicated queries if made from the same IP address in a short period of time [22].

Similar search methods have been previously used in a study on searches related to antibiotics and probiotics [23]. We recorded the data from 1 January 2004 to the date of collection (22 July 2019). We created jointly a list of different body locations and factors related to pain based on general clinical knowledge and main symptoms described in the Polish version of the manual “The Patient History: Evidence-Based Approach” of Tierney et al. [24]. The initial list of chosen characteristics of the pain is presented in Appendix A. We typed the term “pain” and chosen body locations or factors into GT to choose suitable topics and GT matched as topics a total of 24 terms related to pain locations (“Abdominal pain,” “Back pain,” “Breast pain,” “Chest pain,” “Dysmenorrhea,” “Dyspareunia,” “Ear pain,” “Epigastric pain,” “Eye pain,” “Groin pain,” “Headache,” “Knee pain,” “Low back pain,” “Neck pain,” “Odynophagia,” “Pelvic pain,” “Penile pain,” “Podalgia,” “Rectal pain,” “Shoulder pain,” “Sore throat,” “Testicular pain,” “Toothache,” and “Wrist pain”). Some of the topic names are medical terms. Nevertheless, GT assigns colloquialisms to them; for instance, “pain swallowing” was assigned to the topic “Odynophagia.” We did not take the general topic “Pain” into account. We typed all chosen topics separately (non-adjusted data) as well as in comparison with the topic “Abdominal pain” (adjusted data). We chose “Abdominal pain” as the benchmark due to its high prevalence. Only two topics were compared simultaneously. We collected data of interest over time and by region. The countries with low search volumes were excluded using GT configuration. Data points were captured at the monthly level. We used a modified version of the protocol by Nuti et al. to report the search conditions and inputs in detail (Appendix A) [21].

### 2.2. Data Processing and Statistical Analysis

To fully analyze the data, we assigned a value of “0.5” to all RSVs described by GT as <1% and “0.1” to all RSVs equaling “0.” The adjusted data of interest over time was used to calculate the mean proportion of the adjusted RSVs of all pain-related topics with reference to “Abdominal pain” (Appendix A). For the topic “Abdominal pain” itself, this proportion was 1.00.

The adjusted data for comparisons by region represents the proportion of RSVs for both topics in a specific country (Appendix A). The sum of both RSVs in a given region equals 100. This approach allows a comparison of queries related to the chosen topics that are more often typed in a specific country. We established the most frequent pain-related topics for all countries. Since the popularity of the topics was always adjusted with reference to the topic “Ab dominal pain,” we set the RSV of “Abdominal pain” in all countries to 50.

We used the non-adjusted interest by region data to identify countries with the highest RSVs for a specific complaint (Appendix A). In this case, RSV = 100 in the analyzed period represents a country with the highest number of queries related to the chosen topic. 

Non-adjusted data for interest over time were used for time series analysis (Appendix A). We performed the Seasonal Mann–Kendall test using R 3.6.1 (R Foundation, Vienna, Austria) *Kendall* package version 2.2 to detect the presence of a significant secular trend in time series data [25]. P-values below 0.05 were considered to indicate significant differences. In cases showing a significant secular trend, we performed a linear regression to estimate the slope expressed as changes in RSV per year. To investigate significant seasonal variations, we fitted to the time trend an exponential smoothing state space model with Box-Cox transformation, autoregressive-moving average errors, trend and seasonal components (TBATS) by using the *forecast* package version 8.9 of R [26]. We extracted the seasonal component by using the Seasonal Decomposition of Time Series by Loess (Local Polynomial Regression Fitting). We also calculated yearly amplitude as the difference between maximal and minimal seasonal components of time series.

## 3. Results

### 3.1. Global Ranking for the Most Common Pain Locations

Globally, Google users most commonly searched for information associated with (data expressed with reference to “Abdominal pain”) “Headache” (1.30), “Abdominal pain” (1.00), “Back pain” (0.84), “Sore throat” (0.46), and “Low back pain” (0.39) (Table 1). We visualized the prevalence of the pain-related topics location in Figure 1.

### 3.2. Local Patterns for the Most Common Pain Locations

From 250 regions, GT reported low search volumes for 180 countries, which were excluded. We visualized the most frequently searched pain-related topics across all countries in Figure 2A,B. For most countries, Google users were particularly interested in “Headache” (n = 41 countries, including most of North and South America, North European countries, Turkey, South Africa, India, Japan, Australia, and New Zealand), “Abdominal pain” (n = 27, including North Africa, East Europe, Near East, Central Asian countries, China, France, and Spain), “Back pain,” and “Neck pain” (both n = 1). The five most common pain-related topics in each specific country are presented in Appendix A.

For each topic, we identified the five countries with the highest RSVs worldwide (Table 2). Seven topics were most often searched by Iranian users while five were most often searched by Vietnamese Google users. The five countries with the highest interest in the topics “Abdominal pain” and “Low back pain” were Asian countries, while the corresponding countries for “Odynophagia” were South American.

### 3.3. Time Series Analysis

We found that interest in all topics significantly (*p* < 0.001) increased over time, except “Dyspareunia,” which showed a decrease in interest (Figure 3, Table 3). The most dynamic increase over time was observed for the topics “Abdominal pain” (5.67 RSV/year), “Toothache” (5.52 RSV/year), and “Groin pain” (5.12 RSV/year). Significant 12-month seasonal variations (*p* < 0.001) were found in all topics except “Epigastric pain,” and “Penile pain.” Seven topic peaks were noted in January, four in February, three each in March and August, two each in May and April, and one in December. Thirteen topics had the lowest RSVs in December, three in September, three in June, and one each in July, August, and November. The highest yearly amplitude was noted for the topics “Sore throat” (RSV, 14.89), “Odynophagia” (RSV, 12.71), and “Dyspareunia” (RSV, 11.39), while the lowest amplitudes were observed for “Toothache” (4.45 RSV), “Neck pain” (4.77 RSV), and “Testicular pain” (4.88 RSV). 

## 4. Discussion

Infodemiology is a novel approach to investigate important health issues and create the background for further studies. The available GT data revealed that globally, Google users are mostly concerned with Headache, Abdominal pain, and Back pain. The interest in pain-related topics was dependent on the country of the individual.

### 4.1. Global Interest 

Headache was the main concern among Google users. Indeed, 46% of people experience at least one episode of Headache during a year [27], and this complaint might be the most common symptom in primary care [28,29].

Abdominal pain, which is reported by 2.8% of patients in primary care, was the second most common complaint after Headache [30]. In a recent large population-based study that combined digital health technology with questionnaires, Almatio et al. found that GI symptoms were highly prevalent among the 71,000 surveyed US citizens. Nearly 25% of the responders had experienced Abdominal pain in the past week [31]. In contrast to Headache (migraine), Back pain, and Neck pain, the causes of Abdominal pain are not in the top ten main causes of years of life in disability [4]. Moreover, among the twenty most common reasons for visiting doctors in the US, esophageal disorders were 12th and bowel disorders were 16th, while back problems were 3rd, upper respiratory diseases (Sore throat, Ear pain) were 5th, and Headache was 10th [32]. These discrepancies may be explained by the fact that Google queries may be typed by users suffering from serious or mild pain that does not require doctor consultation. Another explanation might be that Abdominal pain is frequently accompanied by symptoms that could be perceived as embarrassing, such as flatulence or problems with defecation. For this reason, the Internet and digital tools seem to be attractive alternatives to medical offices. Indeed, a substantial number of patients with functional gastrointestinal disorders (FGIDs) [33] do not seek medical help. Another concern is that primary providers might not deliver the best solutions to patients with certain pain problems, while access to specialist care in many countries is limited [34]. Internet-based health services may therefore bridge the gaps in access to specialist care in remote and underserved areas [35].

Back, Low back, Neck, and Knee pains were the third-, fifth-, eighth-, and tenth-most frequent concerns of Google users. Epidemiological studies have shown that low back, hip, neck, shoulder, and Knee pain are among the most frequently reported complaints and one of the leading problems causing disability and the reasons for seeking medical help [36]. However, this problem is more prevalent among older individuals, and there are no definite data clarifying whether elderly patients are underrepresented among Google users.

Chest pain is the primary complaint in 1–3% of patients in primary care [37], but among Google users, the complaint was three times less frequently searched for than Abdominal pain. This difference may also be related to the subjective severity of the pain and underlying conditions. Podalgia, which showed similar popularity as Chest pain, is reported to occur in 14% of adolescents [38] and over 30% of people aged over 65 years [39]. Foot pain is generally not related to life-threatening conditions, but it may bother a substantial number of Google users; thus, the problem should not be overlooked. In contrast, Toothache affects 35% of the global population, but the actual prevalence is dependent on local dental care [40]. Although Google may be used as an alternative for limited access to the dentist, this requires further investigations.

### 4.2. Regional Interest

The geographical distribution of the most frequently searched pain-related topics in the investigated regions has not been elucidated to date. Indeed, the highest prevalence of Headache was reported to be in North America and the lowest in Africa [41], although there is a lack of studies presenting worldwide comparisons of the prevalence of pain in different locations. Most previous studies focused on chronic pain [4] or specific conditions responsible for a given ailment [29]. Google users may seek remedies for acute, chronic, transient, and mild pain that does not require consultation and are thus not noted in medical records. We assume that geographical differences may be associated with environmental factors such as diet, lifestyle, genetic background, disease spectrum, and climate [42]. Although we presented the top five pain-related topics for each country with high search volumes, the interpretation of these regional rankings would require profound data of the possible regional causes and is beyond the scope of this paper. 

### 4.3. Time Course

In the analyzed years, the consumption of opioid and non-opioid analgesics tends to increase [43,44,45,46]. RSVs of almost all investigated topics increased over time. This phenomenon cannot be explained by the simultaneous increase in Internet users alone because RSV is adjusted to the number of current Google users. We assume that the increase is somehow “natural” due to improvements in health-related websites and expansion of their numbers and content. Moreover, the Internet has become more accessible with the increasing use of mobile devices, which can help older people seek health-related content on the Web [47]. It may be assumed that broader access to the Internet enabled more seniors to seek for health-related information. Therefore, the number of Google pain-related searches might have increased mostly due to elders, who suffer from chronic pain-related conditions associated with aging [48]. Finally, as mentioned previously, limited access to health care [49] or an increase in consultation or procedure waiting times [50] may lead people to seek advice from the Web. 

Importantly, the graph of RSVs over time for topics with lowest interest, such as “Rectal pain” or “Penile pain” appeared irregular (Figure 3.). In contrast, the time series of interest for the most popular pain-related topics such as “Abdominal pain” or “Chest pain” were smoother, with more visible seasonality. However, the decrease in interest in “Dyspareunia” is puzzling. Since the relative interest was low, the search volume could be highly sensitive to irregular variations and underrepresents the actual seasonal trend. Nevertheless, to our knowledge, there are no studies assessing the long-term prevalence of Dyspareunia to confirm this trend. However, the Google Trends engine did match search terms such as “intercourse pain” and “painful sex” to the topic “Dyspareunia”. Therefore, the trend should not be explained by the highly specialized term “Dyspareunia”.

The majority of Google users live in the northern hemisphere; thus, seasonal variations in the time series mainly reflected this population. The peaks of interest in pain-related topics reflected previously reported relationships of pain location and the season for Headaches [51], viral infection of the upper airways (Ear pain, Sore throat), heartburn [52] or angina pectoris (Chest pain) [53], Abdominal pain [54,55], Low back pain [56], sexual activity (Dyspareunia) [57], and injuries (limb pain) [58]. Interestingly, the lowest RSVs for most of the topics were observed in December, which may be associated with the seasonal holidays, when people’s attention could be away from the Internet. The knowledge on seasonal variation of the searches may be useful to target health e-campaigns for users suffering from specific pain.

This is the first study ranking pain-related topics to reflect the main complaints among Internet users. Due to the popularity of the Google search engine, these data represent a massive number of queries. We found that Google users’ interest in pain-related information increased during the years 2004–2019. This increase may be associated with a growing interest in self-education [59] and self-management [60], limited access to health care [61], or an improvement in e-health content, which encouraged users to consult the Web. This phenomenon should encourage professionals to play vital roles in building e-health resources to provide reliable information and promote health [62]. We presented the secular and seasonal trends in these search queries, which revealed some interesting dynamics. We assume that the use of GT may reveal under-researched epidemiological patterns. However, these findings should be verified in real-world studies. We also hope that this infodemiology approach would allow identification of the needs of users in pain and help to accelerate development of new areas in pain-related investigation [63].

## 5. Limitations

The study has several limitations. First, we found only a limited number of topics related to body pain location. GT did not propose topics related to several rare body pain locations such as the palm, ankle, or arm. Furthermore, within searches on Abdominal pain location, pain in the lumbar and hypochondrium regions did not appear as a topic. Second, the analysis included only Google search engine queries, and the percentage of Internet users who use this engine varies between countries. For instance, over 90% of Europeans use Google as the main search engine, while approximately 80–85% of those in the United States use Google [16]. Therefore, these findings may be less representative for some regions. Third, GT allows simultaneous comparisons of only up to five search terms. Consequently, we could only compare the relative interest in the topics. Fourth, GT does not provide any information regarding the gender, age, socio-economic status, occupation, lifestyle, and psychosocial variables of the users. Previous studies have suggested that young people and women were more eager to seek health-related information on the Web [64,65]. Therefore, it is likely that the users’ sex and age might not reflect the real-world population distribution. However, due to a lack of data, we could not verify this hypothesis. 

## 6. Conclusions

Among pain-related topics, “Headache,” “Abdominal pain,” and “Back pain” are the topics of highest interest among Google users. The interest in topics associated with pain increased over time, and the most dynamic increase noted for “Abdominal pain.” GT is a novel tool that allows retrospective investigation of complaints among Internet users. These findings may provide a foundation for future epidemiological investigations. 

## Figures and Tables

**Figure 1 ijerph-17-00954-f001:**
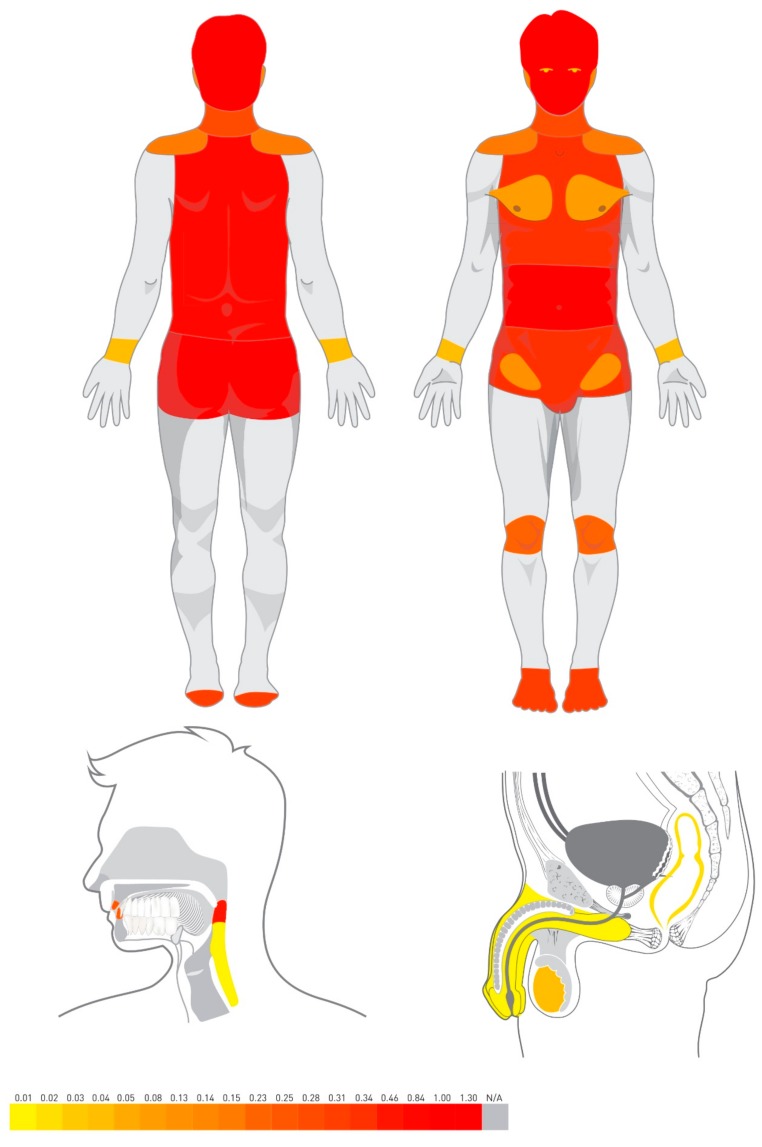
Visualization of the worldwide relative search volume of pain-related topics location.

**Figure 2 ijerph-17-00954-f002:**
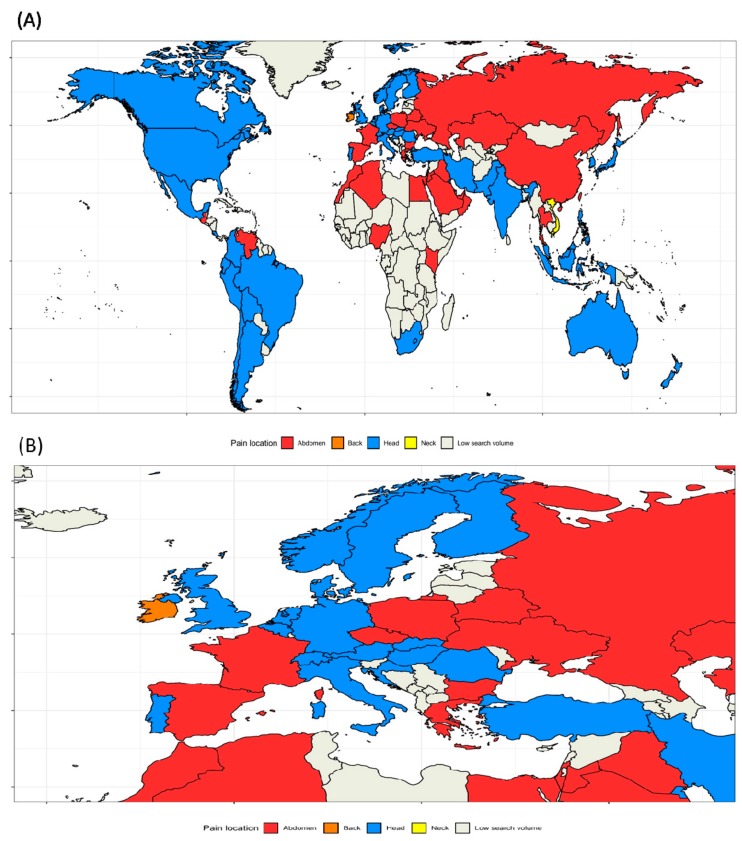
(**A**) World map showing the countries with the highest interest in the pain-related topics (**B**) Europe map showing the countries with the highest interest in pain-related topics.

**Figure 3 ijerph-17-00954-f003:**
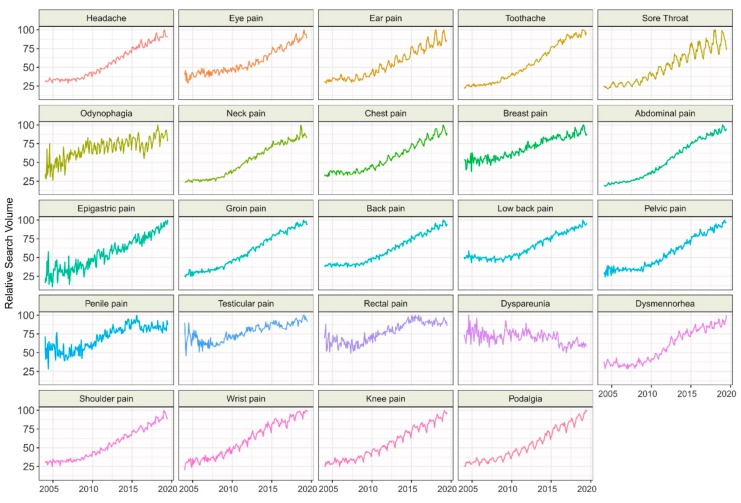
Time trends for relative search volumes of non-adjusted pain-related topics.

**Table 1 ijerph-17-00954-t001:** Popularity of pain-related topics in proportion to “Abdominal pain” (adjusted data; relative search volume [RSV] over time).

Rank	Topic	Proportion of Mean RSV to Abdominal Pain
1.	Headache	1.30
2.	Abdominal pain	1.00
3.	Back pain	0.84
4.	Sore throat	0.46
5.	Low back pain	0.39
6.	Chest pain	0.34
7.	Podalgia	0.31
8.	Neck pain	0.28
9.	Toothache	0.25
10.	Knee pain	0.23
11.	Dysmenorrhea	0.17
12.	Shoulder pain	0.15
13.	Ear pain	0.14
14.	Groin pain	0.13
15.	Breast pain	0.08
16.	Pelvic pain	0.05
17.	Testicular pain	0.04
17.	Wrist pain	0.04
19.	Eye pain	0.03
20.	Dyspareunia	0.02
20.	Rectal pain	0.02
22.	Epigastric pain	0.01
22.	Odynophagia	0.01
22.	Penile pain	0.01

RSV—relative search volume.

**Table 2 ijerph-17-00954-t002:** Mean relative search volume (RSV) of five countries with the highest non-adjusted RSVs for a complaint.

Pain-Related Topic	Five Countries with The Highest RSV of the Topic	
**Head and Neck**		
Headache	Japan (100), Indonesia (97), Iran (79), Saudi Arabia (70), and South Africa (64)	
Eye pain	Iran (100), Taiwan (92), Hong Kong (67), Turkey (65), and Slovakia (63)	
Ear pain	Vietnam (100), Netherlands (78), Germany (67), Brazil (64), and United Kingdom (63)	
Toothache	Indonesia (100), Philippines (36), Turkey (35), Malaysia (34), and Bosnia and Herzegovina (27)	
Sore throat	Iran (100), Singapore (94), Taiwan (82), United States (78), and Ireland (76)	
Odynophagia	Chile (100), Bolivia (84), Honduras (76), Ecuador (69), and Paraguay (69)	
Neck pain	Vietnam (100), Czechia (13), Greece (7), Japan (7), and Romania (7)	
**Trunk**		
Chest pain	Iran (100), Russia (58), Vietnam (58), Belarus (57), and Taiwan (57)	
Breast pain	Jamaica (100), Kenya (97), Nigeria (97), Oman (93), and Trinidad and Tobago (83)	
Abdominal pain	Vietnam (100), Saudi Arabia (69), Japan (67), Oman (66), and Indonesia (62)	
Epigastric pain	Indonesia (100), Vietnam (92), Taiwan (70), Poland (36), and Hong Kong (22)	
Groin pain	Turkey (100), Kenya (55), United Kingdom (46), Ireland (44), and South Africa (38)	
Back pain	Ireland (100), United Kingdom (91), Kenya (90), South Africa (82), and United States (82)	
Low back pain	Japan (100), Iran (77), Indonesia (38), Saudi Arabia (35), and Jordan (33)	
**Pelvic region**		
Pelvic pain	Vietnam (100), Thailand (83), Puerto Rico (28), Kenya (20), and Ireland (18)	
Penile pain	Vietnam (100), Ghana (21), Kenya (16), Singapore (14), and India (13)	
Testicular pain	Iran (100), Poland (35), Czechia (31), Lebanon (31), and Kenya (26)	
Rectal pain	Iran (100), Kenya (44), Poland (44), Sudan (40), and United States (37)	
Dyspareunia	Ghana (100), Japan (75), Kenya (75), Netherlands (75), and Belgium (66)	
Dysmenorrhea	Japan (100), Philippines (88), Jamaica (84), Ghana (77), and Sweden (76)	
**Limbs**		
Shoulder pain	Ireland (100), United Kingdom (93), United States (83), Australia (71), and Canada (67)	
Wrist pain	Iran (100), Netherlands (60), United States (57), United Kingdom (56), and Ireland (55)	
Knee pain	Iran (100), Japan (69), United States (66), United Kingdom (62), and Ireland (60)	
Podalgia	Iran (100), Vietnam (75), Brazil (62), Ireland (57), and United Kingdom (55)	

RSV—relative search volume.

**Table 3 ijerph-17-00954-t003:** Time series analysis of non-adjusted pain-related topics.

Pain-Related Topic	Seasonal Mann-Kendall Test	Slope [RSV/Year]	TBATS (Seasonality Present, Period [Month])	Month with the Highest Seasonal Component [RSV]	Month with the Lowest Seasonal Component [RSV]	Seasonal Component Amplitude [RSV]
Head and Neck						
Headache	tau = 0.96 ***	4.62 ***	YES, 12	February [2.41]	December [−3.61]	6.02
Eye pain	tau = 0.91 ***	3.66 ***	YES, 12	February [2.75]	June [−2.81]	5.55
Ear pain	tau = 0.94 ***	4.23 ***	YES, 12	February [6.63]	September [−4.50]	11.13
Toothache	tau = 0.98 ***	5.52 ***	YES, 12	March [1.98]	June [−2.46]	4.45
Sore throat	tau = 0.98 ***	4.72 ***	YES, 12	December [6.94]	August [−7.96]	14.89
Odynophagia	tau = 0.72 ***	2.58 ***	YES, 12	April [5.24]	July [−7.47]	12.71
Neck pain	tau = 0.97 ***	4.83 ***	YES, 12	March [2.15]	December [−2.62]	4.77
Trunk						
Chest pain	tau = 0.93 ***	4.22 ***	YES, 12	January [4.06]	June [−3.59]	7.65
Breast pain	tau = 0.90 ***	3.05 ***	YES, 12	January [4.93]	December [−4.35]	9.27
Abdominal pain	tau = 1.00 ***	5.67; ***	YES, 12	January [1.93]	November [−1.26]	3.19
Epigastric pain	tau = 0.86 ***	4.55 ***	NO, -	-	-	-
Groin pain	tau = 0.98 ***	5.12 ***	YES, 12	January [1.41]	December [−3.21]	4.63
Back pain	tau = 0.94 ***	4.15 ***	YES, 12	January [2.62]	December [−3.08]	5.70
Low back pain	tau = 0.84 ***	3.40 ***	YES, 12	August [1.88]	December [−4.77]	6.65
Pelvic region						
Pelvic pain	tau = 0.92 ***	4.77 ***	YES, 12	March [1.47]	December [−4.2]	5.67
Penile pain	tau = 0.69 ***	3.00 ***	NO, -	-	-	-
Testicular pain	tau = 0.74 ***	2.19 ***	YES, 12	January [2.30]	September [−2.58]	4.88
Rectal pain	tau = 0.69 ***	2.55 ***	YES, 12	February [2.53]	September [−4.09]	6.63
Dyspareunia	tau = −0.39 ***	-1.00 ***	YES, 12	May [4.18]	December [−7.21]	11.39
Dysmenorrhea	tau = 0.90 ***	4.67 ***	YES, 12	July [4.57]	December [−5.02]	9.59
Limbs						
Shoulder pain	tau = 0.97 ***	4.49 ***	YES, 12	January [3.43]	December [−3.04]	6.47
Wrist pain	tau = 0.96 ***	5.05 ***	YES, 12	August [3.18]	December [−7.20]	10.38
Knee pain	tau = 0.97 ***	4.69 ***	YES, 12	May [3.81]	December [−7.36]	11.16
Podalgia	tau = 0.98 ***	4.63 ***	YES, 12	August [3.94]	December [−6.39]	10.33

*** *p* < 0.001, RSV—relative search volume.

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
