# Peer review of "“Dr. Google, I am in Pain”—Global Internet Searches Associated with Pain: A Retrospective Analysis of Google Trends Data"

_ijerph, 2020, doi:10.3390/ijerph17030954_

Round 1
Reviewer 1 Report
The topic is interesting.
I have the following comments and suggestions:
It seems that the authors only concerned about how frequent the people has searched for their pain, but how they deal with, and whether they have visited the Dr afterwards are unknown. What is the meaning behind if we only know headache and abdominal pain are two most frequent pain that the people searched in google? Do you count those users who repeatedly searched in google? Will you double count the frequency? There are different types of people searching for pain, i.e. student, people in pain, health care provider, relatives of people in pain etc. Can you differentiate them by using the current data? After reading the manuscript, I cannot figure out the purpose of this study. Why do we need to know how many people has searched for pain in google? What is the implication?Author Response
"Please see the attachment."

Reviewer 2 Report
“Dr. Google, I am in pain”—Global Internet searches associated with pain: A retrospective analysis of Google Trends data
This paper explores the relative volume of Google search terms in relation to pain, considering variation across countries and over time. Long term trends and seasonality pattern are examined, and the most prevalent pain-related topics are identified for many countries. It adds to the growing literature on infodemiology which seek to understand previously unobserved behaviours in relation to healthcare.
While the authors are rightly cautious in over-interpreting their results, their observations suggest fundamental questions for policy makers, which are not always as explicit as they might be. For example, does the internet support or undermine frontline healthcare providers? Do search results direct suffers to appropriate websites? Are pain suffers making appropriate decisions in relation to self-treatment versus referral?
Can the dramatic rise in pain related searches (3 or 4-fold) be placed in greater context? For example, in relation to changes in national spend on OTC or prescription pain medication?
Line 19-20: Suggest rewording to “to identify and analyze global interest in topics (n = 24) related to locations of pain and how these progressed over time.”
Line 28-30: This repeats information already provided.
Line 55: “analysis of the Internet data” remove “the”
Line 82-87: This part is unclear. Please provide more detail to aid replicability. How did you select the body locations? Were the topics all suggested by Google? How did you determine which to include and which to omit to arrive at 24?
Line 90: “as well as in”
Line 92: It is worth noting that Google’s market share will be smaller in some countries, and its data may be distorted by other factors over the full time period (e.g. China).
Line 92-93: Clarify that excluding ‘low search volume’ countries is a Google Trend configuration option and therefore does not rely on judgment.
Line 106: “in a given country” to “in all countries”
Line 107-109. Please clarify if this was an author calculation or directly provided by Google Trends.
Line 170-3. This introduction would benefit from discussion of reasons why people might search and how this might affect relative search interest. For example, diagnosis (what is this pain), reassurance (is this normal/serious, is it just me), treatment (self-manageable? options for immediate relief).
Line 187-88. “may be associated with both serious and mild pain that does not require doctor consultation” – should be reworded to clarify the intended meaning and made explicit. Presumably, serious pain may warrant urgent investigation but would ultimately benefit from medical expertise.
Line 200-1. Internet access should act as a reasonable proxy and data is widely available. For example in the UK:
https://www.ons.gov.uk/peoplepopulationandcommunity/householdcharacteristics/homeinternetandsocialmediausage/bulletins/internetaccesshouseholdsandindividuals/2018
The change shown here from 2012-18 may well be part of the story why relative search interest in health-related issues has increased over the same period.
Line 227. I think it’s likely to be more than just content availability. Internet technologies and social media have permeated all aspects of modern life to the extent where more people will find it ‘natural’ to seek (and volunteer) healthcare advice using online sources.
Line 242-3. An implication is that the data you present is more likely to under (rather than over) represent the actual seasonal tend.
Line 242-8. What are the practical implications? For example, could seasonality data be used to time public health campaigns?
Line 304. References are numbered twice.
Author Response
"Please see the attachment."

Reviewer 3 Report
I suggest that the relevance of the study should be specified more clearly in the introduction. In Materials and Methods authors should specify that in Google Trends when searching in English is equivalent to have the option of all languages activated. In figures 1, 2a, 2b and 3 authors should increase the text of descriptions, since they are illegible.Author Response
"Please see the attachment."

Reviewer 4 Report
Thank you for the opportunity to review this interesting paper. It is well written, and does not try
The analysis of internet based data does open an interesting possible for exploratory analysis and hypothesis generation. However, it is important to understand the limitations of the data sources being used.
As noted by the authors it is unclear who 'global Google users' actually are. Results are likely to be weighted towards regions with large populations of Google users, and there is little data on the demographics of this user group.
There is another important limitation that needs to be acknowledged.
Each Google Trends result is based on a sample of the data, and thus results are subject to variability. Results can therefore never be completely reproducible. For example, I reran the search topic 'Epigastric pain' and obtained not only different results from what is presented in the manuscript, but also different results when I reran the search two minutes later. Unfortunately, as the absolute sample sizes are unknown, the level of variability is also unknown. It needs to be noted what is presented in the paper is just one 'sample' of the data, and that different samples may produce different results. To make the results more robust, the authors would need to take multiple 'samples' of the data (ie, refresh their search queries multiple times) and present the results along with measures of variability (eg. standard deviations, standard errors, or 95% confidence intervals).
Regional interest
Differences by region may also be partially explained by language differences. In particular the way the Google maps search terms to topics.
Time course
Changes in RSV over time may be due to changing demographics in the 'global Google users', in particular age.The authors note that the increase in RSV levels for pain may be related to the number of pain-related searchers among elders. It should be noted that the world population, and most likely the age distribution of a global Google user, has increased substantially since 2004. I am not a pain specialist, but it appears to me that whilst the prevalence of many types of pain increases with age, the prevalence of dyspareunia does not.
As noted, seasonal variation are likely driven by the larger proportion of users in the Northern hemisphere. Differences may be larger if it were possible to analyse Northern and Southern hemisphere's independently.
Author Response
"Please see the attachment."

Round 2
Reviewer 1 Report
I appreciate the reponses from the authors, though I am still not convinced with what the study has brought about.